# LEAST PROBABLE DISAGREEMENT REGION FOR ACTIVE LEARNING

## ABSTRACT

Active learning strategy to query unlabeled samples nearer the estimated decision boundary at each step has been known to be effective when the distance from the sample data to the decision boundary can be explicitly evaluated; however, in numerous cases in machine learning, especially when it involves deep learning, conventional distance such as the $\ell_p$ from sample to decision boundary is not readily measurable. This paper defines a theoretical distance of unlabeled sample to the decision boundary as the least probable disagreement region (LPDR) containing the unlabeled sample, and it discusses how this theoretical distance can be empirically evaluated with a lower order of time complexity. Monte Carlo sampling of the hypothesis is performed in approximating the theoretically defined distance. Experimental results on various datasets show that the proposed algorithm consistently outperforms all other high performing uncertainty based active learning algorithms and leads to state-of-the-art active learning performance on CIFAR10, CIFAR100, Tiny ImageNet and Food101 datasets. Only the proposed algorithm outperforms random sampling on CIFAR100 dataset using K-CNN while all other algorithms fail to do so.

## 1 INTRODUCTION

Active learning (Cohn et al., 1996) is a subfield of machine learning to attain data efficiency with fewer labeled training data when it is allowed to choose the training data from which to learn. For many real-world learning problems, large collections of unlabeled samples is assumed available, and based on a certain query strategy, the label of the most informative data is iteratively queried to an oracle to be used in retraining the model (Bouneffouf et al., 2014; Roy & McCallum, 2001; Sener & Savarese, 2017b; Settles et al., 2008; Sinha et al., 2019; Sener & Savarese, 2017a; Pinsler et al., 2019; Shi & Yu, 2019; Gudovskiy et al., 2020). Active learning attempts to achieve high accuracy using as few labeled samples as possible (Settles, 2009).

Of the possible query strategies, uncertainty-based sampling (Culotta & McCallum, 2005; Scheffer et al., 2001; Mussmann & Liang, 2018), which enhances the current model by labeling unlabeled samples that are difficult for the model to predict, is a simple strategy commonly used in pool-based active learning (Lewis & Gale, 1994). Nevertheless, many existing uncertainty-based algorithms have their own limitations. Entropy (Shannon, 1948) based uncertainty sampling can query unlabeled samples near the decision boundary for binary classification, but it does not perform well in multiclass classification as entropy does not equate well with the distance to a complex decision boundary (Joshi et al., 2009). Another approach based on MC-dropout sampling (Gal et al., 2017) which uses a mutual information based BALD (Houlsby et al., 2011) as an uncertainty measure identifies unlabeled samples that are individually informative. This approach, however, is not necessarily informative when it is jointly considered with other samples for label acquisition. To address this problem, BatchBALD (Kirsch et al., 2019) is introduced. However, BatchBALD computes, theoretically, all possible joint mutual information of batch, and is infeasible for large query size. The ensemble method (Beluch et al., 2018), one of the query by committee (QBC) algorithm (Seung et al., 1992), has been shown to perform well in many cases. The fundamental premise behind the QBC is minimizing the version space (Mitchell, 1982), which is the set of hypotheses that are consistent with labeled samples. However, the ensemble method requires high computation load because all networks that make up the ensemble must be trained.

This paper defines a theoretical distance referred to as the least probable disagreement region (LPDR) from sample to the estimated decision boundary, and in each step of active learning, labels of unlabeled samples nearest to the decision boundary in terms of LPDR are obtained to be used for retraining the classifier to improve accuracy of the estimated decision boundary. It is generally understood that labels to samples near the decision boundary are the most informative as the samples are uncertain. Indeed in Balcan et al. (2007), selecting unlabeled samples with the smallest margin to the linear decision boundary and thereby minimal certainty attains exponential improvement over random sampling in terms of sample complexity. In deep learning, it is difficult to identify samples nearest to the decision boundary as sample distance to decision boundary is difficult to evaluate. An adversarial approach (Ducoffe & Precioso, 2018) to approximate the sample distance to decision boundary has been studied but this method does not show preservation of the order of the sample distance and requires considerable computation in obtaining the distance.

## 2    DISTANCE: LEAST PROBABLE DISAGREEMENT REGION (LPDR)

This paper proposes an algorithm for selecting unlabeled data that are close to the decision boundary which can not be explicitly defined in many of cases.

Let $\mathcal{X}$, $\mathcal{Y}$, $\mathcal{H}$ and $\mathcal{D}$ be the instance space, the label space, the set of hypotheses $h : \boldsymbol{x} \rightarrow y$ and the joint distribution over $(\boldsymbol{x}, y) \in \mathcal{X} \times \mathcal{Y}$. The distance between two hypotheses $\hat{h}$ and $h$ is defined as the probability of the disagreement region for $\hat{h}$ and $h$. This distance was originally defined in Hanneke et al. (2014) and Hsu (2010):

$$\rho(\hat{h}, h) := \mathbb{P}_{\mathcal{D}}[\hat{h}(X) \neq h(X)]. \tag{1}$$

This paper defines the sample distance $d$ of $\boldsymbol{x}$ to the hypothesis $\hat{h} \in \mathcal{H}$ based on $\rho$ as the least probable disagreement region (LPDR) that contains $\boldsymbol{x}$:

$$d(\boldsymbol{x}, \hat{h}) := \inf_{h \in \mathcal{H}(\boldsymbol{x}, \hat{h})} \rho(\hat{h}, h) \tag{2}$$

where $\mathcal{H}(\boldsymbol{x}, \hat{h}) = \{h \in \mathcal{H} : \hat{h}(\boldsymbol{x}) \neq h(\boldsymbol{x})\}$.

Figure 1 shows an example of LPDR. Let's define $\mathcal{H} = \{h_\theta : h_\theta(x) = \mathbb{I}[x > \theta]\}$ on input $x$ sampled from uniform distribution $\mathcal{D} = U[0, 1]$ where $\mathbb{I}[\cdot]$ is an indicator function. Suppose $x = x_0$ and $\hat{h} = h_a \in \mathcal{H}$ when $a < x_0$. Here, $\mathcal{H}(x_0, h_a)$ consists of all hypotheses whose prediction on $x_0$ is in disagreement with $h_a(x_0) = 1$, i.e., $\mathcal{H}(x_0, h_a) = \{h_b \in \mathcal{H} : h_b(x_0) = 0\} = \{h_b \in \mathcal{H} : b > x_0\}$. Then, the LPDR between $x_0$ and $h_a$, $d(x_0, h_a) = x_0 - a$ as the infimum of the distance between $h_a$ and $h_b \in \mathcal{H}(x_0, h_a)$ is $\rho(h_a, h_{x_0}) = x_0 - a$.

Figure 1: An example of LPDR between a sample $x = x_0$ and a hypothesis $\hat{h} = h_a$ in binary classification using the $h_\theta(x) = \mathbb{I}[x > \theta]$ on input $x \sim U[0, 1]$.

Here, the sample distribution $\mathcal{D}$ is unknown, and $\mathcal{H}(\boldsymbol{x}, \hat{h})$ may be uncountably infinite. Therefore, a systematic and empirical method for evaluating the distance is required. One might the procedure below: Sample hypotheses sets $\mathcal{H}' = \{h' : \rho(\hat{h}, h') \leq \rho'\}$ in terms of $\rho'$, and perform grid search to determine the smallest $\rho'$ such that there exists $h' \in \mathcal{H}'$ satisfying $\hat{h}(\boldsymbol{x}) \neq h'(\boldsymbol{x})$ for a given $\boldsymbol{x}$. Sampling the hypotheses within the ball can be performed by sampling the corresponding parameters with the assumption that the expected hypothesis distance is monotonically increasing for the expected distance between the corresponding parameters (see Assumption 1). This scheme is based on performing grid search on $\rho'$ and is therefore computationally inefficient. However, unlabeled samples can be ordered according to $d$ without grid search with the assumption that there exists a $\mathcal{H}'$ such that variation ratio $V(\boldsymbol{x}) = 1 - f_m^{(\boldsymbol{x})}/|\mathcal{H}'|$ and $d(\boldsymbol{x}, \hat{h})$ have strong negative correlation where $f_m^{(\boldsymbol{x})} = \max_c \sum_{h' \in \mathcal{H}'} \mathbb{I}[h'(\boldsymbol{x}) = c]$ (see Assumption 2).

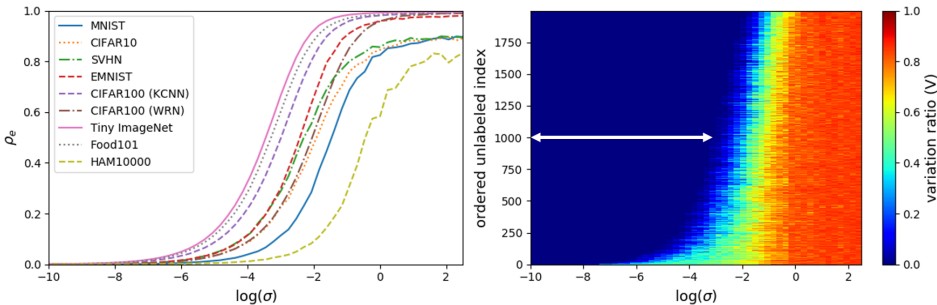

Figure 2: Empirical validation of Assumption 1. Left figure: Relationship between approximated hypothesis distance and $\sigma$ at step $t = 0$. Hypothesis distance is almost linearly proportional to $\log(\sigma)$ in the ascension. Right figure: Relationship between variation ratio and $\sigma$ (MNIST). Sample distance to the decision boundary can be expressed as $\sigma$ at which the variation ratio is not zero for the first time (white arrow). The unlabeled samples are ordered in terms of LPDR.

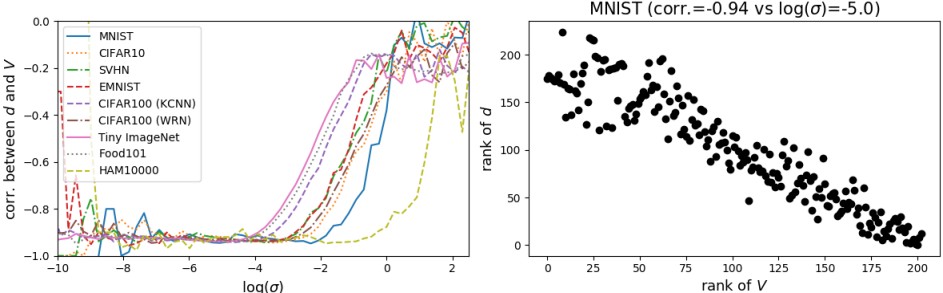

Figure 3: Empirical validation of Assumption 2. Left figure: Spearman's rank correlation coefficient between LPDR and the variation ratio in terms of $\sigma$ showing that there exists a $\sigma$ such that LPDR and the variation ratio have a strong rank correlation. Right figure: An example of strong negative correlation between both ranks when $\log(\sigma) = -5.0$. Samples with increasing LPDR or variation ratio are ranked from high to low.

**Assumption 1.** *The expected distance between $\hat{h}$ and randomly sampled $h$ is monotonically increasing in the expected distance between the corresponding $\hat{\boldsymbol{w}}$ and $\boldsymbol{w}$, i.e., $\mathbb{E}[\|\hat{\boldsymbol{w}} - \boldsymbol{w}_1\| \mid \hat{\boldsymbol{w}}] \leq \mathbb{E}[\|\hat{\boldsymbol{w}} - \boldsymbol{w}_2\| \mid \hat{\boldsymbol{w}}]$ implies that $\mathbb{E}[\rho(\hat{h}, h_1) \mid \hat{h}] \leq \mathbb{E}[\rho(\hat{h}, h_2) \mid \hat{h}]$ where $\hat{\boldsymbol{w}}, \boldsymbol{w}_1$ and $\boldsymbol{w}_2$ are the parameters pertaining to $\hat{h}, h_1$ and $h_2$ respectively.*

**Assumption 2.** *There exists a hypothesis set $\mathcal{H}'$ sampled around $\hat{h}$ having the property that large variation ratio for a given sample data implies small sample distance to $\hat{h}$ with high probability, i.e., there exists $\mathcal{H}'$ such that $V(\boldsymbol{x}_1) \geq V(\boldsymbol{x}_2)$ implies that $d(\boldsymbol{x}_1, \hat{h}) \leq d(\boldsymbol{x}_2, \hat{h})$ with high probability.*

## 3 EMPIRICAL STUDIES OF LPDR

### 3.1 HYPOTHESES AND PARAMETERS IN DEEP NETWORKS: ASSUMPTION 1

The distance between two hypotheses can be approximated by vectors of the predicted labels on random samples by the hypotheses:

$$\rho(\hat{h}, h) \approx \rho_e(\hat{h}, h) = \frac{1}{m} \sum_{i=1}^{m} \mathbb{I}\left[ \hat{h}(\boldsymbol{x}^{(i)}) \neq h(\boldsymbol{x}^{(i)}) \right] \tag{3}$$

where $\boldsymbol{x}^{(i)}$ is the $i^{\text{th}}$ sample for $i \in [m]$. The $h$ is sampled by sampling model parameter $\boldsymbol{w} \sim \mathcal{N}(\hat{\boldsymbol{w}}, \mathbf{I}\sigma^2)$ where $\hat{\boldsymbol{w}}$ is the model parameter of $\hat{h}$, and the expectation of distances between $\boldsymbol{w}$ and $\hat{\boldsymbol{w}}$ depends on $\sigma$. The $\rho_e$ is obtained by the average of 100 times for a fixed $\sigma$. The left-hand side of Figure 2 shows the relationship between $\rho_e$ and $\sigma$ on various datasets and deep networks. The $\rho_e$ increases almost monotonically as $\sigma$ increases. This implies that the order is preserved between

the $\sigma$ and $\rho_e$. Furthermore, the $\rho_e$ is almost linearly proportional to $\log(\sigma)$ in the ascension of the graph, i.e., $\sigma \propto e^{\beta \rho_e}$ for some $\beta > 0$. The right-hand side of Figure 2 shows $V$ with respect to $\sigma$ for each unlabeled sample on MNIST. The sample distance to the decision boundary can be expressed as $\sigma$ at which the variation ratio is not zero for the first time (white arrow), where the indices of unlabeled samples in y-axis are ordered by LPDR. The variation ratio increases as the $\sigma$ increases, and it is expected that the data point with short distance has the large variation ratio compared to the data point with long distance on a certain range of $\sigma$.

## 3.2 LPDR vs Variation Ratio: Assumption 2

The left-hand side of Figure 3 shows Spearman's rank correlation coefficient (Spearman, 1904) between LPDR and the variation ratio with respect to $\sigma$. The correlation is calculated using only unlabeled samples whose variation ratio is not $0$. The strong rank correlation is verified when the $\sigma$ has the appropriate value. Too larger value of $\sigma$ generates hypotheses too far away from $\hat{h}$, which is not helpful to measure the distance. The right-hand side of Figure 3 shows an example of $\sigma$ ($\log(\sigma) = -5.0$) which makes LPDR and the variation ratio have a strong negative correlation on MNIST, that is, the data point with larger variation is closer to the decision boundary. Results for various datasets and networks are presented in Appendix C.

The time complexity is discussed to validate the efficiency of using variation ratio. Let $m$, $N$ and $n_\sigma$ be the unlabeled sample size, $|\mathcal{H}'|$ and the number of grid for $\sigma$ respectively. Ordering unlabeled samples in terms of LPDR by grid search with respect to $\sigma$ requires the time complexity of $m \times N \times n_\sigma$ (see the right-hand side of Figure 2). However, using variation ratio for ordering unlabeled samples reduces the time complexity to $m \times N$. In the case of $n_\sigma = cN$ for some $c > 0$, then the time complexity can be reduced from $O(mN^2)$ to $O(mN)$.

# 4 Algorithm for LPDR

## 4.1 Framework

Let $\mathcal{L}_t$ and $\mathcal{U}_t$ be the labeled and unlabeled samples at step $t$. At step $t$, LPDR trains model parameters $\hat{w}_t$ using labeled samples $\mathcal{L}_t$, and constructs $\mathcal{H}'$ by sampling the model parameters $w'_n \sim \mathcal{N}(\hat{w}_t, \mathbf{I}\sigma^2)$ for $n \in [N]$. Then, LPDR queries the top $q$ unlabeled samples having highest variation ratio from the pool data $\mathcal{P}_t \subset \mathcal{U}_t$ of size $m$.

## 4.2 Construction of Sampled Hypothesis Set

It is important to set an appropriate $\sigma$ when constructing $\mathcal{H}'$ as variation ratios goes to $0$ with decreasing $\sigma$ (see the right-hand side of Figure 2) and the rank correlation goes to zero with increasing $\sigma$ (see the left-hand side of Figure 3). Theoretically, let's consider the binary classification with logistic regression where the predicted label is defined as $y = \text{sgn}(\boldsymbol{x}^\mathsf{T} \boldsymbol{w})$ and $\sup_{\boldsymbol{x} \in \mathcal{X}} \|\boldsymbol{x}\|_\infty < \infty$. Then the following theorem holds and the proof is described in Appendix A.

**Theorem 1.** *Suppose that $w'_n$ for $n = 1, \ldots, N$ are generated with the variance of $\sigma^2$. For all $\boldsymbol{x}$, the followings hold: 1) As $N \to \infty$, $1 - f_m^{(\boldsymbol{x})}/N$ goes to $0$ in probability as $\sigma^2$ goes to $0$, 2) As $N \to \infty$, $1 - f_m^{(\boldsymbol{x})}/N$ goes to $1/2$ in probability for binary classification using logistic regression as $\sigma^2$ goes to $\infty$.*

The implication of Theorem 1 is that when $\sigma$ is too small or too large, it would be difficult to compare the sample distances of unlabeled samples. In this active learning task, at least $q$ most informative unlabeled samples must be identified. To meet this condition, it is reasonable to set $\rho'_n$ denoted in the Algorithm 1 as $\rho^* = q/m$, which is not very small and is less than $1/2$ in general, for $N$ hypotheses.

This can be attained by updating $\sigma'_n$ as $\sigma'_{n+1} = \sigma'_n e^{-\beta(\rho'_n - \rho^*)}$ where $\beta > 0$ (see Appendix D). The Figure 10 in Appendix E shows the final test accuracy with respect to target $\rho'_n$ on MNIST dataset. The LPDR performs best when the target $\rho'_n$ is roughly $\rho^*$. In addition, the range of target $\rho'_n$, associated with the best performance, is wide; thus, LPDR is relatively robust against target $\rho'_n$. Furthermore, LPDR is robust against hyperparameters $\beta$, $N$ and sampling layers (see Appendix F).

---

**Algorithm 1** Least Probable Disagree Region (LPDR)

---

**Input**:
$\mathcal{L}_0, \mathcal{U}_0$ : Initial labeled and unlabeled samples
$m, q$ : Size of pool data and number of queries
$\sigma_0^2$ : Initial variance for sampling
$\rho^*$ : Target hypothesis distance $(= q/m)$

**Procedure:**
 1: **for** step $t = 0, 1, 2, \ldots, T - 1$
 2:     Train parameters $\hat{\boldsymbol{w}}_t$ with $\mathcal{L}_t$, then evaluate its empirical error $\hat{\varepsilon}_t$ on $\mathcal{L}_t$
 3:     $\sigma_t \rightarrow \sigma_1'$
 4:     **for** $n = 1, 2, \ldots, N$
 5:         Sample parameters $\boldsymbol{w}_n' \sim \mathcal{N}(\hat{\boldsymbol{w}}_t, \mathbf{I}\sigma_n'^2)$ for $h_n'$
 6:         Compute $\gamma_n = e^{-(\varepsilon_n' - \hat{\varepsilon}_t)_+}$ where $\varepsilon_n'$ is empirical error of $\boldsymbol{w}_n'$ on $\mathcal{L}_t$
 7:         Compute $\rho_n' = \rho_e(\hat{h}_t, h_n')$
 8:         Update $\sigma_{n+1}' = \sigma_n' e^{-\beta(\rho_n' - \rho^*)}$ where $\beta > 0$
 9:     **end for**
10:     $\sigma_{N+1}' \rightarrow \sigma_{t+1}$
11:     Compute $V_w(\boldsymbol{x}^{(i)}) = 1 - f_w^{(i)} / \sum_{n=1}^N \gamma_n$ where $f_m^{(i)} = \max_c \sum_{n=1}^N \gamma_n \mathbb{I}\left[h_n'(\boldsymbol{x}^{(i)}) = c\right]$
12:     Get $\mathcal{I}^* = \arg \max_{\mathcal{I} \subset \mathcal{I}_{\mathcal{P}_t}, |\mathcal{I}|=q} \sum_{i \in \mathcal{I}} V_w(\boldsymbol{x}^{(i)})$ where $\mathcal{I}_{\mathcal{P}_t} = \left\{ j : \boldsymbol{x}^{(j)} \in \mathcal{P}_t \subseteq \mathcal{U}_t \right\}$
13:     Update $\mathcal{L}_{t+1} = \mathcal{L}_t \cup \left\{ \left(\boldsymbol{x}^{(i)}, y^{(i)}\right) \right\}_{i \in \mathcal{I}^*}$ and $\mathcal{U}_{t+1} = \mathcal{U}_t \setminus \left\{ \boldsymbol{x}^{(i)} \right\}_{i \in \mathcal{I}^*}$
14: **end for**

---

Meanwhile, the efficiency of querying samples in the disagreement region of the version space is well known both theoretically (Hanneke et al., 2014) and empirically (Beluch et al., 2018). When the trained hypothesis $\hat{h}_t$ is in the version space, the sampled hypotheses $h_n'$s are in the version space with high probability, but there are cases where they are outside the version space (see Appendix G).

Thus, LPDR gives weight $\gamma_n$ on the prediction of sampled hypothesis $h_n'$ where $\gamma_n = e^{-\left(\varepsilon_n' - \hat{\varepsilon}_t\right)_+}$ is a function of $\hat{\varepsilon}_t = \mathrm{err}_{\mathcal{L}_t}(\hat{h}_t)$ and $\varepsilon_n' = \mathrm{err}_{\mathcal{L}_t}(h_n')$. Here, $(\cdot)_+$ is $\max\{0, \cdot\}$ and $\mathrm{err}_{\mathcal{L}}(h)$ is the empirical error of $h$ on $\mathcal{L}$. Then, LPDR uses weighted variation ratio $V_w$ as a function of the weighted frequency of the modal class $f_w$ as defined below:

$$V_w(\boldsymbol{x}^{(i)}) = 1 - \frac{f_w^{(i)}}{\sum_{n=1}^N \gamma_n} \tag{4}$$

where $f_w^{(i)} = \max_c \sum_{n=1}^N \gamma_n \mathbb{I}\left[h_n'(\boldsymbol{x}^{(i)}) = c\right]$ and $\boldsymbol{x}^{(i)} \in \mathcal{P}_t \subseteq \mathcal{U}_t$.

If $\mathcal{H}'$ is a subset of the version space in realizable case, the sample complexity of LPDR follows Hanneke's theorem (Hanneke et al., 2014). Let $\Lambda$ be the sample complexity defined as the smallest integer $t$ such that for all $t' \geq t$, $\mathrm{err}(h_{t'}) \leq \epsilon$ where $\mathrm{err}(h) := \mathbb{P}_{\mathcal{D}}[h(X) \neq Y]$ with probability at least $1 - \delta$. Then, LPDR achieves a sample complexity $\Lambda$ such that, for $\mathcal{D}$ in the realizable case, $\forall \epsilon, \delta \in (0, 1)$,

$$\Lambda(\epsilon, \delta, \mathcal{D}) \lesssim \xi \cdot \left( D \log \xi + \log \left( \frac{\log(1/\epsilon)}{\delta} \right) \right) \cdot \log \frac{1}{\epsilon}$$

where $D$ and $\xi$ are the VC-dimension of $\mathcal{H}$ and the disagree coefficient with respect to $\mathcal{H}$ and $\mathcal{D}$.

When $\xi = O(1)$, in terms of $\epsilon$, the number of labeled samples required by LPDR is just $O(\log(1/\epsilon) \cdot \log \log(1/\epsilon))$, while the number of labeled samples by a passive learning is $\Omega(1/\epsilon)$. Therefore, in this case, LPDR provides an exponential improvement over passive learning in sample complexity (Hsu, 2010).

## 5   EXPERIMENTS

This section discusses experimental results on 8 benchmark datasets: MNIST (LeCun et al., 1998), CIFAR10 (Krizhevsky et al., 2009), SVHN (Netzer et al., 2019), EMNIST (Cohen et al., 2017), CIFAR100 (Krizhevsky et al., 2009), Tiny ImageNet (subset of the ILSVRC dataset containing

Table 1: Experimental settings for comparing the performance on various datasets are summarized. Epochs is the maximum number of training epochs. Data size denotes the sizes of datasets for training / validation / test. Acquisition size denotes the number of samples for the initial model + number of samples acquired in each step (from the number of samples in the pool data) → Maximum number of samples acquired during training.

| Dataset | Model | Epochs | Data size train / val / test | Acquisition size | | |
|---|---|---|---|---|---|---|
| MNIST | S-CNN | 50 | 55,000 / 5,000 / 10,000 | 20 | +20 (2K) | → 1,020 |
| CIFAR10 | K-CNN | 150 | 45,000 / 5,000 / 10,000 | 200 | +100 (2K) | → 10,200 |
| SVHN | K-CNN | 150 | 68,257 / 5,000 / 26,032 | 200 | +100 (2K) | → 10,200 |
| EMNIST | K-CNN | 100 | 75,000 / 5,000 / 10,000 | 235 | +150 (2K) | → 15,235 |
| CIFAR100 | K-CNN | 150 | 45,000 / 5,000 / 10,000 | 2,000 | +500 (5K) | → 27,000 |
| CIFAR100 | WRN-16-8 | 100 | 45,000 / 5,000 / 10,000 | 5,000 | +2,000 (10K) | → 25,000 |
| Tiny ImageNet | WRN-16-8 | 200 | 90,000 / 10,000 / 10,000 | 10,000 | +5,000 (20K) | → 50,000 |
| Food101 | WRN-16-8 | 200 | 60,600 / 15,150 / 25,250 | 6,000 | +3,000 (15K) | → 30,000 |
| HAM10000 | WRN-16-8 | 100 | 7,015 / 1,500 / 1,500 | 500 | +300 (3K) | → 3,500 |

200 categories rather than the usual 1000 categories; Russakovsky et al., 2015), Food101 (Bossard et al., 2014) and HAM10000 (Tschandl et al., 2018) datasets. For fair comparison with other active learning algorithms, simple two layered CNN, referred to as 'S-CNN' (Chollet et al., 2015) is used for MNIST and four layered CNN, referred to as 'K-CNN' (Chollet et al., 2015) is used for CIFAR10, SVHN, EMNIST and CIFAR100. Additionally, Wide-ResNet (WRN-16-8; Zagoruyko & Komodakis, 2016) is used for CIFAR100, Tiny ImageNet, Food101 and HAM10000.

Figure 4–7 show magnified plots of test accuracy to accentuate the difference in performance among different methods: initial labeled sample sizes are not shown in the figures. Figures that include initial labeled sample size are presented in Appendix H.

## 5.1 EXPERIMENTAL SETTINGS

Experimental settings regarding total number of epochs, data size and acquisition size are summarized in Table 1, and other details concerning the model, optimizer, batch size, learning rate and hyperparameters are presented in Appendix B.

## 5.2 RESULTS FOR MNIST, CIFAR10, SVHN AND EMNIST

A number of experiments are conducted to compare performance of LPDR with other high performing uncertainty based active learning algorithms on 8 datasets. Figure 4 shows the test accuracy with respect to the number of labeled samples on MNIST, CIFAR10, SVHN and EMNIST datasets. Each algorithm is denoted such as 'LPDR': the proposed algorithm, 'Random': random sampling, 'Entropy': entropy based uncertainty sampling, 'MC-BALD': MC dropout sampling using BALD, 'MC-VarR': MC dropout sampling using variation ratio (Ducoffe & Precioso, 2015) and 'ENS-VarR': ensemble method. Overall, LPDR either performs best or comparable with all other algorithms. Its performance is consistent regardless of the benchmark datasets. In the early step, LPDR significantly outperforms all other algorithms on MNIST and CIFAR10 datasets. Of all the algorithms compared, Entropy performed the worst. MC-BALD performed well only on SVHN dataset: it seems that the performance of BALD is highly dependent on the dataset. With the query size set to 1, LPDR outperforms BatchBALD on MNIST dataset (see Appendix I). Although MC-VarR and ENS-VarR are based on different sampling methods, both perform similarily-both outperforming all others on EMNIST dataset, while showing a significant drop in performance compared to LPDR on SVHN and CIFAR10 datasets. It is observed that the performances of other algorithms have a relatively strong data dependency compared to LPDR. On CIFAR10 dataset, the performances of MC-VarR and ENS-VarR are no better than that of Random, and Entropy and MC-BALD have lower performance than Random. These results can be attributed to the low network capacity compared to the data complexity. This issue will be discussed in next section.

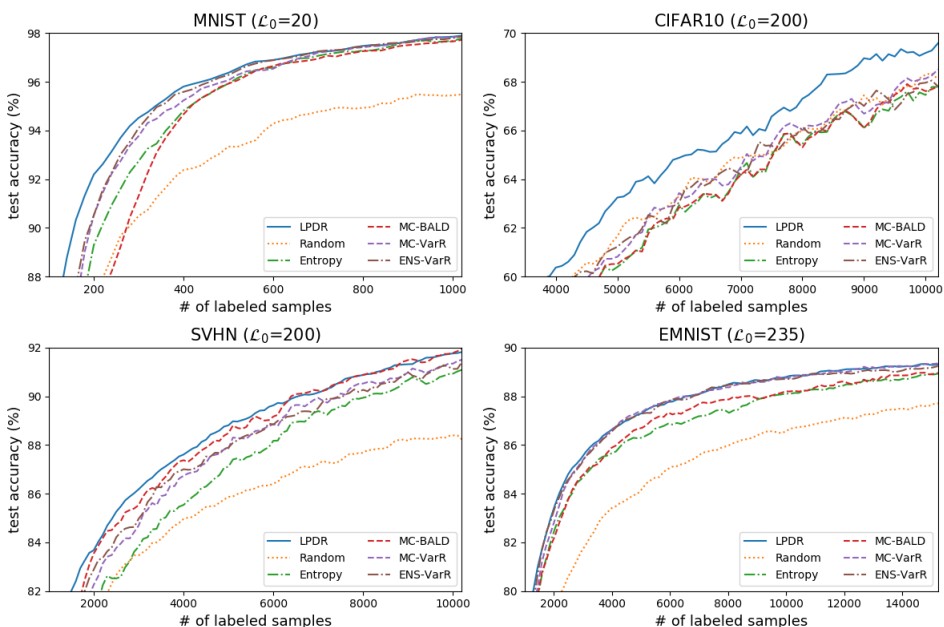

Figure 4: The performance comparison of LPDR with the uncertainty based active learning algorithms on MNIST, CIFAR10, SVHN and EMNIST datasets (Random: random sampling, Entropy: entropy based uncertainty sampling, MC-BALD: MC dropout sampling with BALD, MC-VarR: MC dropout sampling with variation ratio, ENS-VarR: ensemble network with variation ratio). Overall, LPDR consistently either performs best or comparable with all other algorithms regardless of dataset. The performance of all algorithms except LPDR tend to be data dependent.

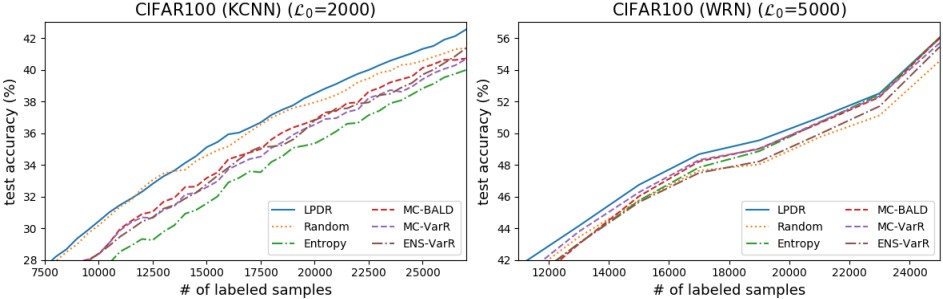

Figure 5: Performance comparison with respect to the network capacity on CIFAR100 dataset. The performances of all algorithms except LPDR are much worse than that of Random when using K-CNN, which has a relatively smaller network capacity than that of WRN-16-8. LPDR is able to perform consistently better than Random regardless of the network capacity.

### 5.3 RESULTS FOR CIFAR100 WITH K-CNN AND WIDE-RESNET

In order to compare the performance of the algorithms with respect to the network capacity, experiments are conducted using networks of different capacity but on the same dataset. Figure 5 shows the results of test accuracy with respect to the number of labeled samples on CIFAR100 dataset with K-CNN and WRN-16-8. The left-hand figure is the results of using K-CNN, which has a relatively smaller network capacity than that of WRN-16-8. With the exception of LPDR, the performances of all algorithms are much worse than that of Random. The right-hand figure is the result of using WRN-16-8, which has a relatively larger network capacity. In contrast to the results for K-CNN, most algorithms outperform Random. With a large network capacity, the performance gap between LPDR and the other algorithms is reduced, but LPDR still outperforms others. LPDR is able to perform consistently better than Random regardless of the network capacity, and it seems to be particularly effective with low capacity networks.

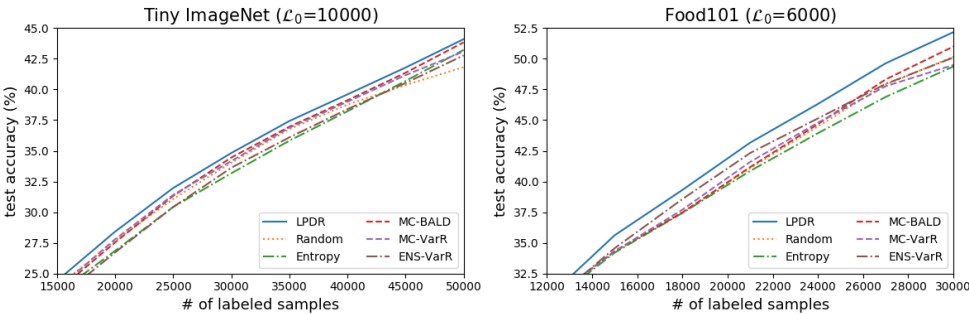

Figure 6: The performance comparison on Tiny ImageNet and Food101 datasets with WRN-16-8. LPDR outperforms all other algorithms in more difficult tasks.

## 5.4 RESULTS FOR TINY IMAGENET AND FOOD101

Experiments on a more difficult task are conducted. Figure 6 shows test accuracy with respect to the number of labeled samples on Tiny ImageNet and Food101 datasets with WRN-16-8. Tiny ImageNet and Food101 are considered to be more difficult than CIFAR100. Even on more difficult tasks, LPDR outperforms all other algorithms.

## 5.5 RESULTS FOR HAM10000

Additional experiments are conducted to compare the performance of the algorithms on imbalanced HAM10000 dataset with WRN-16-8. Figure 7 shows the results of the test accuracy with respect to the number of labeled samples. The LPDR outperforms all other algorithms compared. Figure 15 in Appendix J shows the results of AUC with respect to the number of labeled samples. The LPDR performs comparable with all other algorithms.

To sum up the comparing algorithms across all experimental settings and repetitions, rank and Dolan-More curves are presented in Appendix K. The LPDR consistently achieves top rank for all steps and significantly outperforms the other algorithms in all experimental settings.

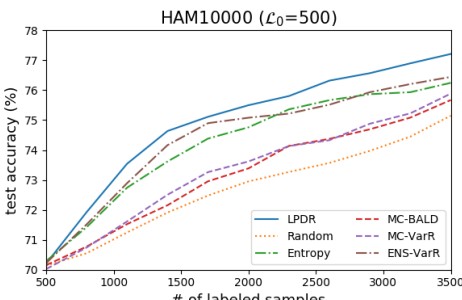

Figure 7: The performance comparison on HAM10000 dataset with WRN-16-8. LPDR outperforms all other algorithms on imbalanced dataset.

## 6 RELATED WORK

Other than uncertainty-based sampling framework (Culotta & McCallum, 2005; Scheffer et al., 2001; Mussmann & Liang, 2018; Lewis & Gale, 1994; Gal et al., 2017; Kirsch et al., 2019; Beluch et al., 2018) for active learning, decision-theoretic framework based methods such as expected model change (Settles et al., 2008) have certain relevance to the proposed LPDR as unlabeled samples nearer the decision boundary which LPDR is attempting to identify have larger gradients leading larger model change. Recently, adversarial approaches are proposed to discriminate labeled and unlabeled samples (Gissin & Shalev-Shwartz, 2019; Sinha et al., 2019; Zhang et al., 2020), and after performing adversarial learning, any unlabeled samples that is most confidently predicted as unlabeled is queried and used to retrain the network. Here adversarial learning is used to indirectly identify sample near the decision boundary.

## 7 CONCLUSION

This paper defines a theoretical distance of unlabeled sample to the decision boundary referred to as the least probable disagreement region (LPDR) containing the unlabeled sample for active learn-

ing. LPDR can be evaluated empirically with low computational load by making two assumptions regarding parameters of the hypothesis space, variation ratio and the LPDR. The two assumptions are empirically verified.

Experimental results on various datasets show that LPDR consistently outperforms all other high performing uncertainty based active learning algorithms and leads to state-of-the-art active learning performance on CIFAR10, CIFAR100, Tiny ImageNet, and Food101 datasets. In addition, LPDR is able to perform consistently better than random sampling regardless of the network capacity while all other algorithms compared fail to do so.

LPDR is simple enough to be applied to various classification tasks with deep networks: the implementation requires only sampling a subset of parameters (parameters in the last FC layer of the deep network). Additionally, LPDR is capable of quick and reliable performance in a variety of different settings with only a computational load that is not much higher than that of other uncertainty sampling methods. In conclusion, LPDR is an effective uncertainty based sampling algorithm in pool-based active learning.

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

APPENDIX

## A    PROOF OF THEOREM 1

Assume that $\|\hat{\boldsymbol{w}}_t\| = 1$ without the loss of generality, and $\|\boldsymbol{x}\| \neq 0$ to avoid the null case. The predicted label of $\boldsymbol{x}$ by $\boldsymbol{w}'_n$ disagrees with that by $\hat{\boldsymbol{w}}_t$ if $\mathrm{sgn}\left(\boldsymbol{x}^\mathsf{T}\boldsymbol{w}'_n\right) \neq \mathrm{sgn}\left(\boldsymbol{x}^\mathsf{T}\hat{\boldsymbol{w}}_t\right)$, here, $\mathrm{sgn}(0) = 1$. Note that

$$\boldsymbol{x}^\mathsf{T}\boldsymbol{w}'_n = \boldsymbol{x}^\mathsf{T}\hat{\boldsymbol{w}}_t + \sigma\boldsymbol{x}^\mathsf{T}\mathbf{e}'_n \quad \text{where} \quad \mathbf{e}'_n = (Z_{n1}, \ldots, Z_{n|\boldsymbol{w}|})^\mathsf{T},$$

and $Z_{nk}$s are independent random variables from $\mathcal{N}(0,1)$. The event of $\{\mathrm{sgn}\left(\boldsymbol{x}^\mathsf{T}\boldsymbol{w}'_n\right) \neq \mathrm{sgn}\left(\boldsymbol{x}^\mathsf{T}\hat{\boldsymbol{w}}_t\right)\}$ is equal to that of $\mathcal{E}_1 \cup \mathcal{E}_2$ where $\mathcal{E}_2 = \{\sigma\boldsymbol{x}^\mathsf{T}\mathbf{e}'_n \geq 0, \boldsymbol{x}^\mathsf{T}\hat{\boldsymbol{w}}_t < 0\}$ and $\mathcal{E}_2 = \{\sigma\boldsymbol{x}^\mathsf{T}\mathbf{e}'_n < 0, \boldsymbol{x}^\mathsf{T}\hat{\boldsymbol{w}}_t \geq 0\}$. Thus, the proof has two folds: the cases of 1) $\mathcal{E}_1$ and 2) $\mathcal{E}_2$.

In the first fold,

$$\mathbb{P}[\mathcal{E}_1] \;=\; \mathbb{P}\left[\sigma\boldsymbol{x}^\mathsf{T}\mathbf{e}'_n \geq |\boldsymbol{x}^\mathsf{T}\hat{\boldsymbol{w}}_t|\right] = \mathbb{P}\left[\sigma\|\boldsymbol{x}\|Z \geq |\boldsymbol{x}^\mathsf{T}\hat{\boldsymbol{w}}_t|\right] = 1 - \Phi\left(\frac{a\left(\boldsymbol{x}, \hat{\boldsymbol{w}}_t\right)}{\sigma}\right)$$

where $Z \sim \mathcal{N}(0,1)$, $\Phi$ is the cumulative distribution function of the normal distribution, and $a\left(\boldsymbol{x}, \hat{\boldsymbol{w}}_t\right) = |\boldsymbol{x}^\mathsf{T}\hat{\boldsymbol{w}}_t|/\|\boldsymbol{x}\|$. Note that $\sigma\boldsymbol{x}^\mathsf{T}\mathbf{e}'_n \sim \mathcal{N}(0, \sigma^2\|\boldsymbol{x}\|^2)$. Consequently, $\mathbb{P}[\mathcal{E}_1] < 1/2$ due to $a\left(\boldsymbol{x}, \hat{\boldsymbol{w}}_t\right) > 0$. Hence, the following

$$\frac{f_m^{(\boldsymbol{x})}}{N} = \sum_{n=1}^{N} \frac{1}{N}\mathbb{I}\left[\hat{h}_t(\boldsymbol{x}) = h'_n(\boldsymbol{x})\right]$$

goes to value greater than 1/2 in probability as $N \to \infty$ because $\mathrm{Var}(f_m^{(\boldsymbol{x})}/N) \to 0$ as $N \to \infty$. Therefore, as $N \to \infty$, $\forall \boldsymbol{x}$, the variation ratio is

$$1 - \frac{f_m^{(\boldsymbol{x})}}{N} = 1 - \sum_{n=1}^{N} \frac{1}{N}\mathbb{I}\left[\hat{h}_t(\boldsymbol{x}) = h'_n(\boldsymbol{x})\right] \to 1 - \Phi\left(\frac{a\left(\boldsymbol{x}, \hat{\boldsymbol{w}}_t\right)}{\sigma}\right)$$

in probability. This is due to that $f_m^{(\boldsymbol{x})}$ is the frequency of mode class with probability tending to 1 as $N \to \infty$. By the smoothness of $\Phi$,

$$1 - \frac{f_m^{(\boldsymbol{x})}}{N} \to 1 - \Phi(\infty) = 0 \quad \text{as} \quad \sigma^2 \to 0$$

and

$$1 - \frac{f_m^{(\boldsymbol{x})}}{N} \to 1 - \Phi(0) = \frac{1}{2} \quad \text{as} \quad \sigma^2 \to \infty.$$

Next, in the second fold,

$$\mathbb{P}[\mathcal{E}_2] \;=\; \mathbb{P}\left[\sigma\boldsymbol{x}^\mathsf{T}\mathbf{e}'_n < -|\boldsymbol{x}^\mathsf{T}\hat{\boldsymbol{w}}_t|\right] = \mathbb{P}\left[\sigma\|\boldsymbol{x}\|Z < -|\boldsymbol{x}^\mathsf{T}\hat{\boldsymbol{w}}_t|\right] = \Phi\left(-\frac{a\left(\boldsymbol{x}, \hat{\boldsymbol{w}}_t\right)}{\sigma}\right).$$

Consequently, $\mathbb{P}[\mathcal{E}_2] < 1/2$. Hence the following

$$\frac{f_m^{(\boldsymbol{x})}}{N} = \sum_{n=1}^{N} \frac{1}{N}\mathbb{I}\left[\hat{h}_t(\boldsymbol{x}) = h'_n(\boldsymbol{x})\right]$$

goes to the value greater than 1/2 in probability as $N \to \infty$ because $\mathrm{Var}(f_m^{(\boldsymbol{x})}/N) \to 0$ as $N \to \infty$. Therefore, as $N \to \infty$, $\forall \boldsymbol{x}$, the variation ratio is

$$1 - \frac{f_m^{(\boldsymbol{x})}}{N} = 1 - \sum_{n=1}^{N} \frac{1}{N}\mathbb{I}\left[\hat{h}_t(\boldsymbol{x}) = h'_n(\boldsymbol{x})\right] \to \Phi\left(\frac{-a\left(\boldsymbol{x}, \hat{\boldsymbol{w}}_t\right)}{\sigma}\right) = 1 - \Phi\left(\frac{a\left(\boldsymbol{x}, \hat{\boldsymbol{w}}_t\right)}{\sigma}\right)$$

in probability. This is due to that $f_m^{(\boldsymbol{x})}$ is the frequency of mode class with probability tending to 1 as $N \to \infty$. By the smoothness of $\Phi$,

$$1 - \frac{f_m^{(\boldsymbol{x})}}{N} \to 1 - \Phi(\infty) = 0 \quad \text{as} \quad \sigma^2 \to 0$$

and

$$1 - \frac{f_m^{(\boldsymbol{x})}}{N} \rightarrow 1 - \Phi(0) = \frac{1}{2} \quad \text{as} \quad \sigma^2 \rightarrow \infty.$$

This completes the proof.

$\square$

## B  EXPERIMENTAL SETTINGS

### B.1  DATASETS

**MNIST** (LeCun et al., 1998) is a dataset of handwritten digits which has a training set of $60,000$ samples and a test set of $10,000$ samples in 10 classes. Each sample is a black and white image and $28 \times 28$ in size.

**CIFAR10** and **CIFAR100** (Krizhevsky et al., 2009) are labeled subsets of the 80 million tiny images dataset which have a training set of $50,000$ samples and a test set of $10,000$ samples in 10 and 100 classes respectively. Each sample is a color image and $32 \times 32$ in size.

**SVHN** (Netzer et al., 2019) is a real-world digit image dataset which has a training set of $73,257$ samples and a test set of $26,032$ samples in 10 classes. Each sample is a color image and $32 \times 32$ in size.

**EMNIST** (Cohen et al., 2017) is a dataset of handwritten character digits which has a training set of $80,000$ samples and a test set of $10,000$ samples in 47 classes. Each sample is a black and white image and $28 \times 28$ in size.

**Tiny ImageNet** is a subset of the ILSVRC (Russakovsky et al., 2015) dataset which has $100,000$ samples in 200 classes. Each sample is a color image and $64 \times 64$ in size. In experiments, Tiny ImageNet is split into two parts: a training set of $90,000$ samples and a test set of $10,000$ samples.

**Food101** (Bossard et al., 2014) is a fine grained dataset which has a training set of $75,750$ samples and a test set of $25,250$ samples in 101 classes. Each sample is a color image and resized to $75 \times 75$.

**HAM10000** (Tschandl et al., 2018) is a imbalanced dataset which has $10,015$ samples in 7 classes. Each sample is a color image and resized to $75 \times 75$. In experiments, HAM10000 is split into two parts: a training set of $8,515$ samples and a test set of $1,500$ samples.

All datasets are used without any preprocessing of images.

### B.2  SETTINGS

S-CNN, which is the Keras MNIST CNN implementation (Chollet et al., 2015), consists of [$3 \times 3 \times 32$ conv - $3 \times 3 \times 64$ conv - $2 \times 2$ maxpool - dropout $(0.25)$ - 128 dense - dropout $(0.5)$ - #class dense - softmax] layers. K-CNN, which is the Keras CIFAR CNN implementation (Chollet et al., 2015), consists of [two $3 \times 3 \times 32$ conv - $2 \times 2$ maxpool - dropout $(0.25)$ - two $3 \times 3 \times 64$ conv - $2 \times 2$ maxpool - dropout $(0.25)$ - 512 dense - dropout $(0.5)$ - #class dense - softmax] layers. WRN-16-8 is a wide residual network that has 16 convolutional layers and a widening factor 8 (Zagoruyko & Komodakis, 2016). The optimizer, initial learning rate, learning rate schedule and batch size for each experimental setting are described in Table 2. He normal initialization is used for all models. All experiments are run for a fixed number of acquisition steps until a certain amount of training data is labeled. Results are averaged over 5 repetitions. For all datasets, the initial labeled samples for each repetition are randomly sampled according to the distribution of the training set. For MC dropout we use 100 forward passes, and ensemble consists of 5 networks of identical architecture but different random initialization and random batches. For LPDR, we set $\sigma_0 = 0.01$, $\beta = 1$, $N = 100$ and parameter sampling is applied to the last dense layer of each network.

Table 2: Settings for Training

| Dataset | Model | Optimizer | Learning Rate | Learning Rate Schedule $\times$decay [epoch schedule] | Batch Size |
|---|---|---|---|---|---|
| MNIST | S-CNN | Adam | 0.001 | - | 32 |
| CIFAR10 | K-CNN | Adam | 0.0001 | - | 64 |
| SVHN | K-CNN | Adam | 0.0001 | - | 64 |
| EMNIST | K-CNN | Adam | 0.0001 | - | 64 |
| CIFAR100 | K-CNN | Adam | 0.0001 | - | 64 |
| CIFAR100 | WRN-16-8 | Nesterov | 0.05 | $\times$0.2 [60, 80] | 128 |
| Tiny ImageNet | WRN-16-8 | Nesterov | 0.1 | $\times$0.2 [60, 120, 160] | 128 |
| Food101 | WRN-16-8 | Nesterov | 0.1 | $\times$0.2 [60, 120, 160] | 128 |
| HAM10000 | WRN-16-8 | Nesterov | 0.05 | $\times$0.2 [60, 80] | 64 |

## C RANK CORRELATION BETWEEN LPDR AND VARIATION RATIO

Figure 8 shows an example of negative Spearman's rank correlation between LPDR and the variation ratio for each experimental setting. Samples with increasing LPDR or variation ratio are ranked from high to low. The $\sigma$ is selected to satisfy $\rho'_n = \rho^* = q/m$ at initial step.

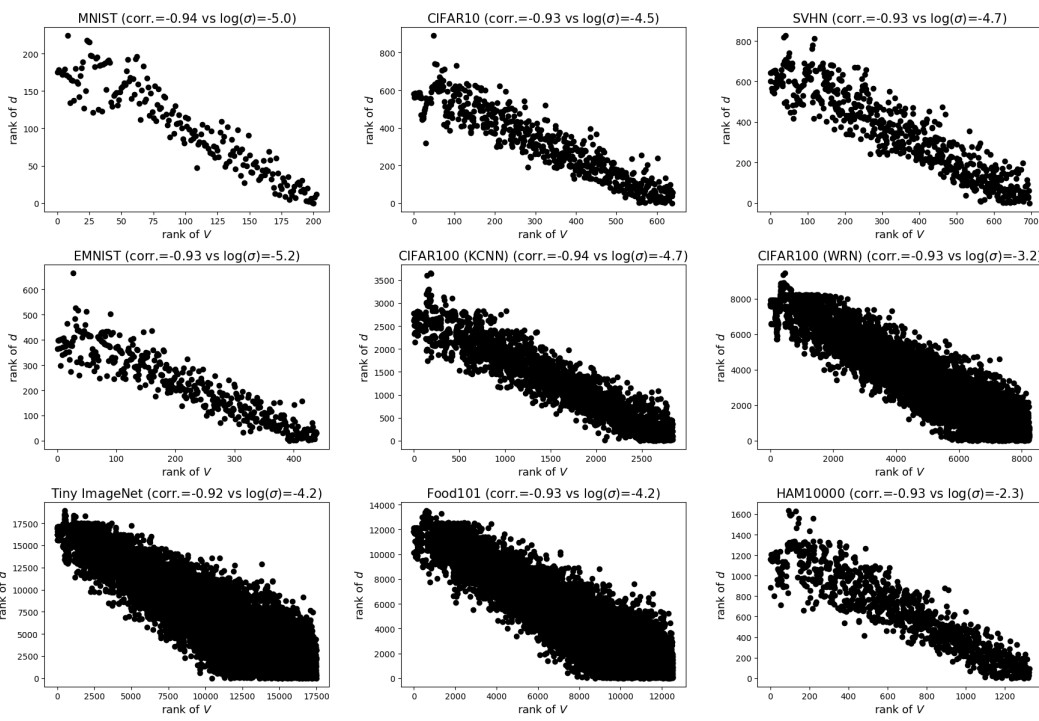

Figure 8: An example of negative Spearman's rank correlation between LPDR and the variation ratio for each experimental setting.

## D REGULATING $\rho'_n$ BY THE VARIANCE OF SAMPLING

The left-hand side of Figure 9 shows the $\rho'_n$ with respect to the active learning progress. For all experiments, LPDR reliably guides the $\rho'_n$ to be $\rho^* = q/m$ (MNIST: 0.01, CIFAR10: 0.05, SVHN: 0.05, EMNIST: 0.075, CIFAR100 (KCNN): 0.1, CIFAR100 (WRN): 0.2, Tiny ImageNet: 0.25, Food101: 0.2 and HAM10000: 0.1) after the initial few steps. The right-hand side of Figure 9 shows

$\log(\sigma)$ with respect to the active learning progress. For all experiments, the variance of sampling increases as the labeling proceeds. This is because larger variance is required to make the $\rho'_n = \rho^*$ since unlabeled samples move away from the learned decision boundary from labeled samples due to an increase in network confidence as the number of labeled samples increases.

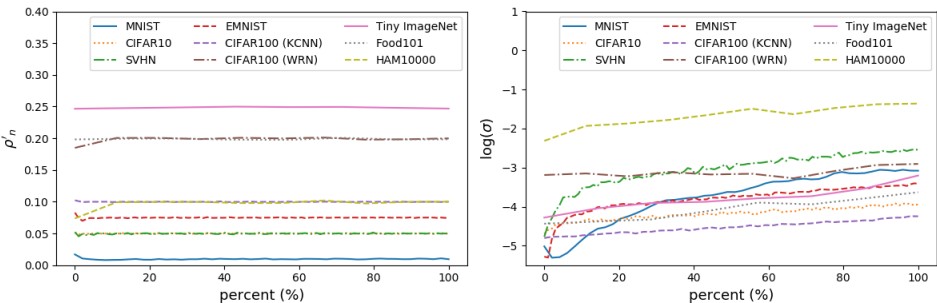

Figure 9: The $\rho'_n$ and $\sigma$ with respect to the labeling proceeds for all experimental settings. LPDR reliably guides the $\rho'_n$ to be the target value by increasing the variance of sampling as the number of labeled samples increases.

## E    FINAL TEST ACCURACY VS TARGET $\rho'_n$

The Figure 10 shows the final test accuracy with respect to target $\rho'_n$ on MNIST dataset. The results show that at around $\rho^*(= 0.02)$, it performs the best for $q = 20$ and $m = 1000$. In addition, the range of target $\rho'_n$, associated with the best performance, is wide $(0.01 \sim 0.1)$; thus, LPDR is robust against the target $\rho'_n$ in the wide range.

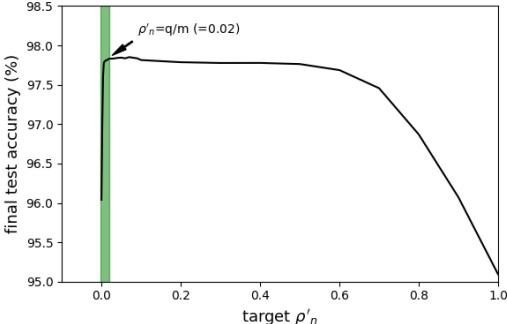

Figure 10: The final accuracy with respect to the target $\rho'_n$ on MNIST dataset. LPDR performs best in a wide range of the target $\rho'_n$.

## F    ROBUSTNESS OF LPDR AGAINST HYPERPARAMETERS

LPDR has four hyperparameters: 1) the initial variance of sampling $\sigma_0$; 2) the positive hyperparameter for regulating the variance of sampling $\beta$; 3) the number of sampled hypotheses $N$, and 4) the layer index of the network to which sampling is applied. The $\sigma_0$ has no significant effect on the performance of LPDR since $\sigma$ is adaptively regulated based on the $\rho'_n$ while sampling the sampled hypothesis. Thus, $\sigma_0$ is not examined in detail. Figure 11 shows the performance comparison with respect to the hyperparameters of LPDR on MNIST and CIFAR10 datasets. The left figures show that there is no significant difference in the performance of LPDR for various $\beta \in \{0.1, 1, 10\}$ on both datasets. The robustness of LPDR against $\beta$ is based on the sufficient buffer for regulating $\sigma$ since the range of target $\rho'_n$ associated with the best performance is wide. The middle figures show that there is no significant difference in the performance of LPDR for various $N \in \{5, 10, 20, 50, 100, 200\}$ on both datasets. The robustness of LPDR against $N$ is based on the sufficient discrimination in the variation ratio for identifying $q$ most informative unlabeled samples with a small number of sampled

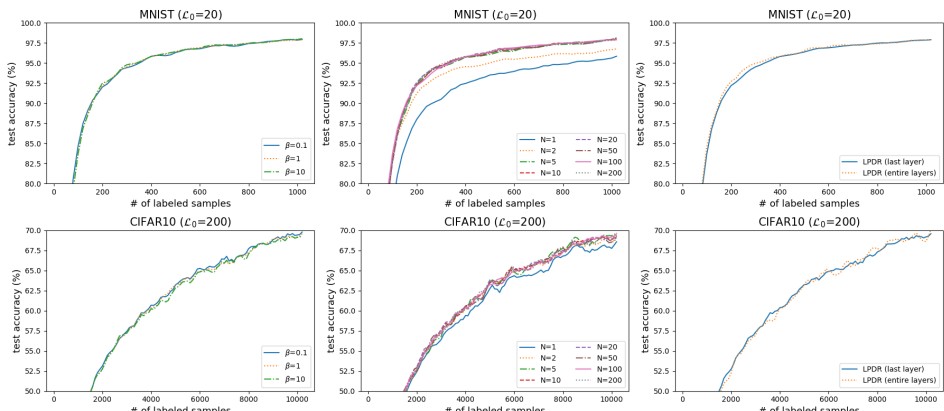

Figure 11: The performance comparison with respect to the hyperparameters of LPDR on MNIST and CIFAR10 datasets. LPDR is robust against $\beta$ and $N$, and has no significant performance difference whether the sampling is applied to the parameters of last layer or all layers.

hypotheses by setting $\rho^* = q/m$. The right figures show that there is no significant difference in the performance of LPDR for the sampling to the parameters of last layer and to the parameters of all layers of the networks on both datasets.

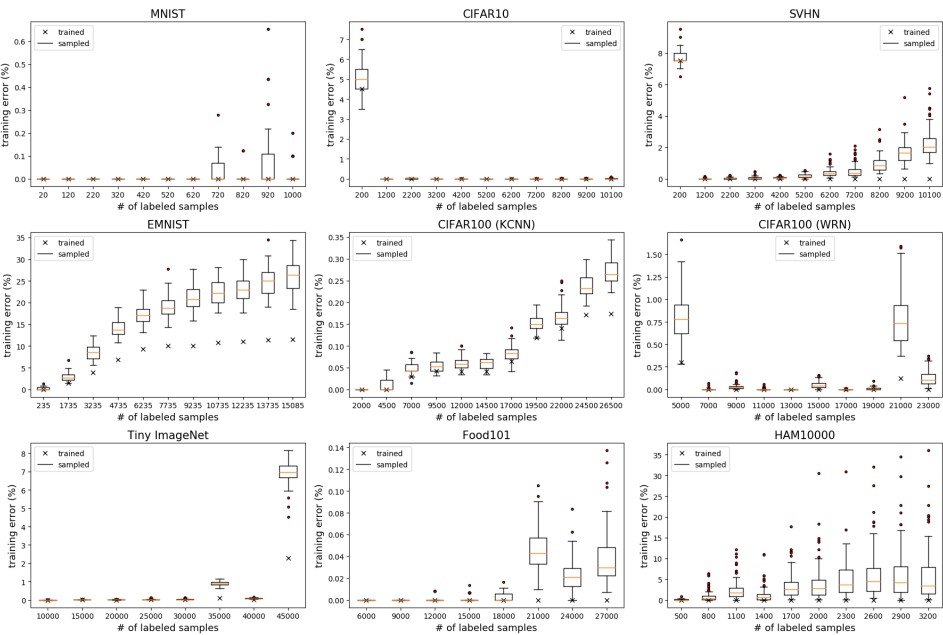

Figure 12: The empirical errors of the learned and the sampled hypotheses with respect to the acquisition step for all experimental settings. It is observed that the empirical error of the learned hypothesis or the sampled hypothesis is not zero.

## G    EMPIRICAL ERRORS OF LEARNED AND SAMPLED HYPOTHESES

Figure 12 shows the empirical error of the learned and the sampled hypotheses with respect to the acquisition step for all experimental settings. In many cases, the empirical error of the learned hypothesis becomes zero, thus it is placed in the version space, while the sampled hypothesis is often placed outside the version space, e.g., in SVHN dataset. Even in the cases of EMNIST and CIFAR100 with K-CNN datasets, as the number of labeled samples increases, the empirical error of the learned and the sampled hypothesis increases. To address this situation, LPDR incorporates the

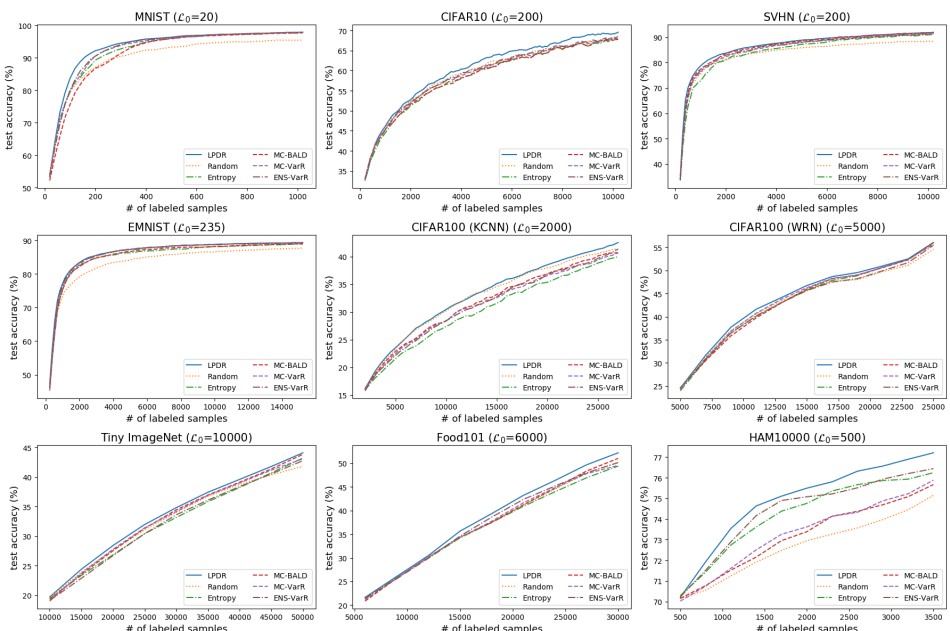

Figure 13: The test accuracy with respect to the number of labeled samples from initial to final step for all experimental settings.

weighted hypotheses based on the prediction error difference between the learned and the sampled hypotheses, and it works well empirically.

## H  PLOTS FOR TEST ACCURACY

Figure 13 shows the test accuracy with respect to the number of labeled samples from initial to final step for all experimental settings.

## I  LPDR VS MC-BATCHBALD

Figure 14 shows the performance comparison between LPDR and MC-BALD on MNIST dataset using S-CNN when the query size is 1 or 20. LPDR significantly outperforms MC-BatchBALD on MNIST dataset when $q = 1$ such that MC-BatchBALD is completely identical to MC-BALD. LPDR is also expected to outperform MC-BatchBALD even when $q > 1$: LPDR with $q > 1$ performs better than MC-BALD with $q = 1$ that MC-BatchBALD with $q > 1$ does not exceed (Kirsch et al., 2019).

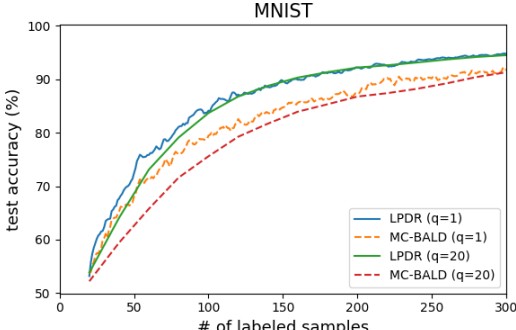

Figure 14: The comparison of performance between LPDR and MC-BALD on MNIST dataset where the query size is 1 or 20. The performance of BatchBALD with $q > 1$ does not exceed that of MC-BALD ($q = 1$) and LPDR ($q = 20$) outperforms MC-BALD ($q = 1$).

## J AUC OF HAM10000 DATASET

On imbalanced dataset, the performance comparison is performed not only for accuracy but also for AUC. Figure 15 shows the results of AUC with respect to the number of labeled samples on HAM10000 dataset. LPDR performs comparable with Entropy or ENS-VarR performing better than other algorithms.

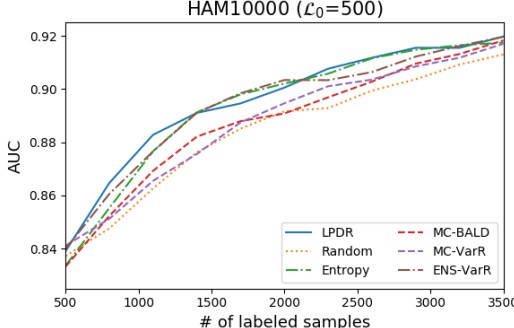

Figure 15: The comparison of AUC on HAM10000 dataset. LPDR performs comparable with the best performing algorithms.

## K RANK AND DOLAN-MORE CURVES

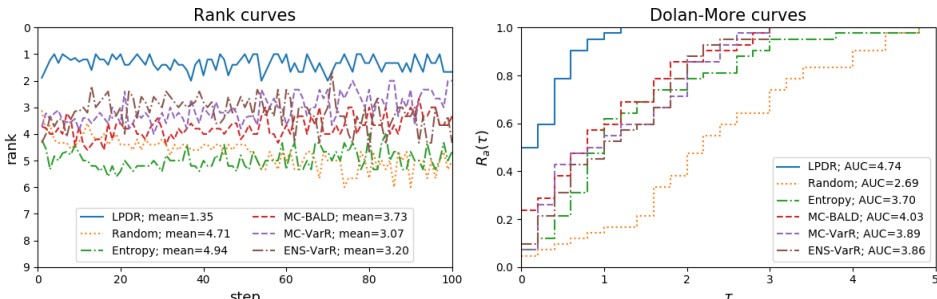

Figure 16: The rank and Dolan-More curves of the algorithms across all experimental settings and repetitions. The left figure shows rank curve which is the mean of ranks on all datasets at each step. LPDR consistently is top-ranked for all steps. The right figure shows the each algorithm's Dolan-More curves which present the fraction of problems in which the algorithm has the performance gap from the best competitor. LPDR maintains the highest value for all $\tau$.

Table 3: The mean ($\pm$ standard deviation) of performance gap from the best competitor for all steps of each algorithm on each dataset. LPDR significantly outperforms the other algorithms on all datasets.

|  | MNIST | CIFAR10 | SVHN | EMNIST | CIFAR100 | CIFAR100-W | T. ImageNet | Food101 | HAM10000 |
|---|---|---|---|---|---|---|---|---|---|
| LPDR | **0.17±0.04** | **0.22±0.07** | **0.15±0.02** | **0.13±0.04** | **0.23±0.10** | **0.50±0.15** | **0.42±0.18** | **0.13±0.09** | **0.29±0.11** |
| Random | 3.45±0.46 | 1.56±0.07 | 3.08±0.16 | 2.80±0.08 | 0.62±0.17 | 1.67±0.46 | 1.41±0.49 | 1.74±0.48 | 2.52±0.39 |
| Entropy | 1.20±0.27 | 2.28±0.19 | 1.92±0.15 | 0.95±0.13 | 3.19±0.34 | 1.19±0.28 | 1.82±0.36 | 2.17±0.03 | 0.98±0.40 |
| MC-BALD | 1.91±0.48 | 2.25±0.16 | 0.45±0.04 | 0.78±0.18 | 1.80±0.35 | 1.17±0.10 | 1.00±0.22 | 1.53±0.09 | 2.04±0.29 |
| MC-VarR | 0.86±0.30 | 1.76±0.46 | 1.06±0.11 | 0.21±0.09 | 2.20±0.44 | 0.92±0.27 | 1.06±0.37 | 1.72±0.01 | 1.92±0.30 |
| ENS-VarR | 0.65±0.34 | 1.74±0.18 | 0.97±0.06 | 0.22±0.04 | 2.13±0.28 | 1.57±0.34 | 1.85±0.26 | 1.31±0.31 | 0.81±0.29 |

Rank curves and Dolan-More curves are used to compare the performance of the algorithms across all experimental settings and repetitions. Figure 16 shows the rank and Dolan-More curves for all

algorithms considered in the experiment. The rank curve of each algorithm in the left-hand figure represents the mean of ranks on all datasets at each steps of active learning. LPDR consistently is top-ranked for all steps.

The right-hand figure shows Dolan-More curves defined as follows (Dolan & Moré, 2002). Let $\mathrm{acc}_a^p$ be the final test accuracy of the $a$ algorithm on the $p$ problem. After defining the performance gap as $\Delta_a^p = \max_x(\mathrm{acc}_x^p) - \mathrm{acc}_a^p$, we can define Dolan-More curve $R_a(\cdot)$ as a function of the performance gap factor $\tau$:

$$R_a(\tau) = \frac{\#(p : \Delta_a^p \leq \tau)}{n_p}$$

where $n_p$ is the total number of evaluations for the problem $p$. Thus, $R_a(\tau)$ is the ratio of problems with performance gap between algorithm $a$ and the best performing competitor not more than $\tau$. Note that $R_a(0)$ is the ratio of problems on which algorithm $a$ performs the best. LPDR has the highest value $R_a(0) = 43.3\%$, and LPDR maintains the highest $R_a(\tau)$ for all $\tau$.

Table 3 presents the mean and the standard deviation of performance gap from the best competitor for all steps of each algorithm on each dataset. Consistent with all the results so far, LPDR significantly outperforms the other algorithms in all experimental settings.

