# OpenReview forum: "Least Probable Disagreement Region for Active Learning"
_ICLR.cc/2021/Conference — Reject_

### Official Review · AnonReviewer2 · 2020-10-20
**A promising method, but more clear analysis is required for some of the claims.**

**Rating:** 5
**Confidence:** 4

**Review:**

The paper defines a new measure of distance between a hypothesis $h$ and a point $x$, which is the probability mass of the smallest (by probability mass) disagreement region (induced by the other $h' \in \mathcal{H}$) containing $x$. In general this is intractable so the authors offer two assumptions about the relationship between this measure and more tractable quantities (one being the distance between the model parameters of those hypotheses, and the other being what the authors call the 'variation ratio'). The reasonability of these assumptions is then assessed, and the algorithm is tested on a variety of dataset and against several reasonable competitors.

Pros:

Theorem 1 provides insight into the behaviour of $f_m^{(x)}$ in the binary linear classifier setting, although it strongly uses the linearity in the proof and so it could not be easily extended beyond that. However its purpose is to justify intuition about the variance of the noise used when sampling perturbed model parameters, rather than to make a strong claim, and it seems suitable for that purpose.

The experimental results are quite promising, with the proposed method outperforming many existing methods for active learning on neural networks. In particular the experiments perform several sets of experiments which give good insight into the general power and robustness of the method (although it would be good to know more about how hyperparameter tuning, in particular was equal computation given to tuning hyperparameters for the competitors)?

Cons:

The paper would benefit from some significant rewriting, as it is quite difficult to follow right now. There are several cases where quantities are used and discussed before being defined (Assumption 2 in the start of section 2) or before being rigorously defined ($f_m^{(x)}$ in Assumption 2 and section 3), or without being defined at all ($\mathcal{P}_t$ in algorithm 1). In Theorem 1 it should be more clear that the theorem only applies to the case of a linear classifier described above it, as right now a brief read through the paper suggests it applies for the neural networks which are used throughout the rest of the paper.

Theorem 2 however is a very strong statement, and there is not sufficient justification for this strong a claim. In particular it is unclear why the authors feel any of these settings are realizable (capable of achieving 0 error on the population from which any of these data samples are drawn from). This is exactly the rate achieved by CAL (see http://www.stevehanneke.com/docs/active-survey/active-survey.pdf page 41), an algorithm which requires search over the entire version space, where as the algorithm here does a very limited search over the version space, otherwise it is at risk of missing whether a point is in the disagreement region. Additionally it is unclear what the authors mean by "When the version space exists...". The justification for the claim here must be made much more explicit.

The algorithm uses several fairly ad hoc choices, such as the use of the weights $\gamma$, which could be much better explained.

Overall I think this method is exciting, but would benefit from additional explanation. And the paper requires further editing and significant expansion around Theorem 2.

---

> ### Author Response · Authors · 2020-11-23
> **Response to AnonReviewer2**
>
> - We have revised the manuscript to the best of our ability to improve readability.
>
> - In the revised manuscript, Assumption 2 and $\mathcal{P}_t$ are used after being defined, and $f_m^{(x)}$ is rigorously defined.
>
> - Theorem 1 has been clarified: we have modified the statement and included phrase "for binary classification using logistic regression".
>
> - We agree with the reviewer's comment that it is a very strong statement: to assume that deep learning for any dataset is realizable would be an exaggeration. Therefore, Theorem 2 is deleted instead a statement is made with regards to the sample complexity of LDPR.
>
> - The weight $\gamma$ is inspired by version space reduction where the version space is reduced by querying samples belonging to the disagreement region of the version space: samples outside the disagreement region of the version space are not queried. If we query a sample using $\mathcal{H}'$ with hypotheses whose training error is not zero, a sample outside the disagreement region of the version space can be queried. To prevent this, $\mathcal{H}'$ should be composed of hypotheses with a training error of zero, and this can be done by rejecting the hypothesis with a training error of non-zero during sampling. However, in practice, not only the sampled hypotheses but also the trained hypothesis may have a non-zero training error. In this case, hard rejection may make it impossible to execute the algorithm itself. For example, in the case of SVHN, when it reaches step of about 40, it fails to sample $N$ hypotheses in 1 million attempts of sampling. Therefore, the penalty is applied in exponential form based on the training error of the trained hypothesis without completely rejecting it.

---

> > ### Comment · AnonReviewer2 · 2020-11-23
> > **Comments**
> >
> > Thank you for responding to my concerns. After the changes I have updated my score.

---

### Official Review · AnonReviewer3 · 2020-10-28
**Shows promising results but messy and unclear presentation and lack of conceptual motivation.**

**Rating:** 4
**Confidence:** 4

**Review:**

This paper defines a new "distance to the decision boundary" and empirically evaluates the corresponding active learning algorithm.

Unfortunately, the motivation for this new distance to the decision boundary is lacking. It seems that there are a variety of possible quantities available and why this new one is principled or where it came from is unclear.

Clarity:
 - There are a lot of quantities, functions, and assumptions floating around and it is not clear how they are motivated or related.
 - I'm assuming $V(x)$ is the variation ratio as defined in Assumption 2? This is not clearly written.
 - In Theorem 1: it appears that $a(x,w)$ is undefined in the proof, or at least is defined after it is used.
 - In Theorem 1: the statement is not self-contained

Correctness:
 - In Theorem 1: it appears that if $x$ is on the decision boundary (i.e. $x^T \hat{w}_t = 0$) then the first case does not hold.

Questions:
 - Why do you want $\ell_p$ distance to the decision boundary instead of uncertainty measures (e.g. entropy)?
 - How is $\beta$ set in the the algorithm? Is this another hyperparameter to tune?


Typo: "least probably disagreement region" in Introduction


***After author response***

Thanks for the response.

Perhaps I didn't explain my concern about the motivation. SVM with uncertainty sampling works well, yes; however, it is equivalently several different things including selecting points closest to the decision boundary in Euclidean distance but also selecting points with highest predictive entropy and selecting points with smallest predictive margin. In other words, it seems that you are extrapolating from the SVM case to the neural network case and it's not clear which of several equivalent things will work in the neural network case.

Furthermore, this paper doesn't even use Euclidean distance but instead uses a newly defined distance. So I really don't find the motivation or theory convincing.

---

> ### Author Response · Authors · 2020-11-23
> **Response to AnonReviewer3**
>
> - As stated in the introduction, for active learning with a hypothesis such as SVM with clearly defined decision boundary such that the distance between the sample and the decision boundary is easily measured, it is known theoretically and empirically that querying the sample close to the decision boundary produces good performance. However, it is difficult to measure the Euclidean distance in deep learning, so we define the distance from sample to the deep network as the distance between the deep network and the closest deep network which the two networks disagree on the prediction of the sample.  The distance ($\rho_e$) can be approximated by Eq(3). To obtain minimum $\rho_e$, we would have to perform grid search increasing the strength of the perturbation from a very small perturbation which the two deep networks agree until we find a  perturbation where the perturbed deep network disagree. This would have to be conducted for each sample point but instead based on Assumption 2 and negative Spearman's rank correlation coefficient between LPDR and the variation ratio, select a perturbation- for instance $\log(\sigma)=-5.0$ - and rank the variation ratio of the entire unlabeled data and select the sample point with the largest variation ratio. Here this sample is assumed to have the smallest LPDR as shown in Figure 3 since correlation between validation ration and LPDR is $-0.9$. As a side note, Assumption 2 is validated by the empirical observation that the correlation of validation ratio and LPDR is negative.
>
> - This paper defines the distance (LPDR, $d$) between the sample and the hypothesis (deep network) using the hypothesis distance ($\rho$), and then introduces assumption 1, 2 and the variation ratio ($V$) to find the order of LPDRs of samples without evaluating the LPDR. Therefore, quantities, functions and assumptions in the paper were introduced as needed, and other quantities and functions for example or description were summarized.
>
> - Yes, $V$ is the variation ratio, and it has a negative Spearman's rank correlation with LPDR. Therefore by Assumption 2 (empirically validated by Figure 3), we can indirectly find the sample with minimum LPDR by ranking the variation ratio of the entire unlabeled data and the sample with largest variation ratio with have the smallest LPDR.
>
> - $a ( x, \hat{w}_t ) = \vert x^T  \hat{w}_t \vert  / \Vert x \Vert$, and we have placed its definition right after its first use. We revised the statement and proof of Theorem 1 by following comments.
>
> - We compare our distance to the entropy based method proposed in "Uncertainty Sampling" (Figure 4,5,6 and 7). We have found our method based on LPDR to be superior to entropy based method. As mentioned in the Introduction, $\ell_p$ distance is difficult to measure with a deep network.
>
> - As stated in Appendix F, $\beta$ is another hyperparameter and there is no significant difference in performance for $\beta$ value for $0.1$, $1$, and $10$. No special tuning is required for $\beta$. Algorithm is insensitive to the value of $\beta$: in the experiment, $\beta=1$.
>
> - Typo is corrected.

---

### Official Review · AnonReviewer1 · 2020-10-28
**Overall, the paper is a solid piece of work. However, there are some questions and minor comments below that could be helpful to improve the quality of the paper.**

**Rating:** 7
**Confidence:** 3

**Review:**

This paper is motivated by the idea that unlabelled samples near the estimated decision boundary show to be very informative/useful in an active learning setting. However, measuring the distance between an instance and the decision boundary is a non-trivial task in numerous machine learning algorithms, especially in deep learning. The paper proposes a (theoretical) sample distance to the decision boundary that relies on the least probable disagreement region (LPDR) that still contains the sample. The paper makes two assumptions to evaluate the proposed distance empirically: (1) closeness of the parameters of two hypotheses implies closeness of these hypotheses as defined by the probability of the disagreement region and (2) the variation ratio of labels obtained by evaluating a set of hypotheses sampled around the decision boundary is a proxy for the proposed distance. Considering these assumptions, hypotheses are sampled around a given decision boundary by adding gaussian noise to the parameters of the fitted model. Both assumptions are validated empirically on different datasets and varying levels of variance of the noise term to show the effect on the variation ratio and distance respectively. Consequently, an iterative active learning algorithm is proposed which adapts the variance of the noise term in order to select the predefined number of samples. Extensive experimental results indicate that LPDR outperforms other uncertainty based active learning algorithms on various datasets or is at least on par with them.

The submission provides a methodological contribution by proposing a way to evaluate the distance of an instance to the decision boundary for deep learning models. Even though the idea of adding noise to the model parameters and obtaining an ensemble of hypotheses is not groundbreaking, it follows a nice intuition and shows good and interesting empirical results. I think the paper is relevant for the ICLR and (active) learning community in general. The submission is well written and good to follow, the approach is well motivated and the concept of using variation ratio as a proxy for the distance is explained in a reasonable way. The paper draws connections to existing active learning approaches and provides numerous empirical results that highlight different aspects of LPDRs performance on various datasets and model architectures. However, it should be noted that the experiments were only conducted on image datasets and corresponding CNN architectures.

The paper includes reasonably complete information about the experiments and models/settings and I think that the reproducibility of the results is good (even though I did not explicitly validate it). However, the authors do not provide source code etc. in order to easily reproduce the results, which I would highly encourage.

Questions / Clarifications / Remarks:

(1) Figure 1 shows the approximated distance between hypothesis h and 100 sampled hypotheses for different levels of sigma. t=0 indicates that the models were trained with a set of samples of initial acquisition size (given by Table 1). I assume that this set was kept constant across the different runs and sigmas. Can you report on the effect of choosing different initial pools? Are the results consistent with other chosen initial pools?
(2) Can you comment on the results in appendix F showing that the performance does not differ significantly if sampling is applied to the parameters of the last or all layers. Do you think there might be scenarios in which this choice matters?
(3) Did you test LPDR on tasks other than image classification and if yes, were the results equally good? If not, do you expect any differences in behaviour?

Minor comments:

(1) Add a description of the white arrow in the right plot of Figure 1.
(2) Please revise the second last sentence on page 3. (The right hand side of ...)
(3) Line 8 and 10 in Algorihm 1: sigma' in the inner loop has index N+1 (line 8) after completing the iteration, but index N is assigned to the sigma in the outer loop.
(4) Please have your submission proof-read for English style and grammar issues such as pronouns.

---

> ### Author Response · Authors · 2020-11-23
> **Response to AnonReviewer1**
>
> - Source code will definitely be made available. For now the code will be made available in the supplementary material.
>
> - The same trend shown in Figure 2 (revised version) can be obtained using different initial pools. The algorithm does not depend on the initial pool since the hypothesis distance between a hypothesis and perturbed hypothesis represented by $\rho_e$ does not depend on the decision boundary and the Hamming distance is approximated by the entire training data rather than the initial labeled pools.
>
> - Consider two extreme cases for perturbation: (1) sampling last layer and (2) sampling entire layer. Parameter sampling only in the last layer detects samples close to the decision boundary as the last layer defines the decision boundary while parameter sampling all the layers perturbs both feature and the decision boundary with a smaller variance (it turns out the perturbation is smaller when perturbing the entire layer in comparison when perturbing the last layer for the same Hamming distance). The samples detected are about the same (comparing last layer perturbation and entire network) as the features are only perturbed slightly in the case for entire layer perturbation and thus samples that are relatively far from decision boundary will stay far and only samples near the decision boundary will be detected with decision boundary which is perturbed slightly. The overall effect is that there seems to be little difference in performance in both cases. Parameter sampling in all layers has the potential to detect samples far from the decision boundary in case the feature changes abnormally, but in general, this is to expected to occur less frequently than detecting samples that are close to the decision boundary.
>
> - We have not conducted extensive experiment dataset other than for image classification. However, we believe that the algorithm can be adapted to work equally well on classification tasks other than images.
>
> - Minor comments: (1) The white arrow denotes the approximation of the LPDR distance in the sigma domain and has been added. (2) The sentence was revised. (3) It was corrected with $\sigma_{N+1}'$. (4) The paper will carefully proof-read the manuscript.

---

### Official Review · AnonReviewer4 · 2020-10-31
**OK but not good enough**

**Rating:** 4
**Confidence:** 2

**Review:**

- Some notions in Section 2 are confusing to me (maybe because I'm not familiar with related literature). For example, I don't understand why Eq.(2) is a kind distance measure between a sample x and a hypothesis.  How to more intuitively understand a "distance of a sample to a hypothesis"?
- The sentence "small perturbation in model parameter leads to small perturbation in hypothesis" does not precisely match the mathematical meaning of Assumption 1.  Assumption 1 is states that $\rho(h,h')$ is an increasing function of $\|w-w'\|^2$.  But the above sentence is better described by $\rho(h,h') < C\|w-w'\|^2$.  Please change either Assumption 1 or the sentence.
- Assumption 1 and 2 are stated in a very strong way. For example, Assumption 1 states that the relation holds for "any" triple of $\hat{w}, w_1, w_2$. However, in Figure 1, the plotted result is "averaged" over random choices of $w_1, w_2$. So Figure 1 does not really verify Assumption 1. Similar for the case of Assumption 2. Please either change the statement of the assumptions or change the experiments in Figure 1 & 2.
- The experiments in Figure 9 is not extensive. To show that target $\rho_n'$ is indeed close to $q/m$, one needs to change the value of $q/m$ and see the corresponding $\rho_n'$.  Also, Figure 9 seems to me "the smaller $\rho_n'$, the better test accuracy".  But in any case, we need more extensive experiments to make any more interpretation.  Besides, do you have any theoretical justification about why $\rho_n'$ should be close to $q/m$?
- The authors hoped to connect their algorithm with some theory, but the assumptions are not stated in a precise way. The "averaged" plots in Figure 1&2 is intuitive in itself, so I don't think it is necessary to create those complicated notions in Section 2.
- The empirical performance of the proposed algorithm is always close to the best baseline, or outperforming it, which is good; but the insensitivity of parameter to the final performance (Section E and F) makes me worry that whether those parameters are taking any effect. Maybe N can be simply chosen as 1 to maximize the computational efficiency? And target $\rho_n'$ can simply set to some very small quantity according to Figure 9 and we don't need to search for it?
- Overall, the proposed algorithm seems to be a heuristic algorithm without precise theoretical justification, and the experiments are not extensive in some aspects.  So I think it should go through a major revision before it can be accepted.

---

> ### Author Response · Authors · 2020-11-23
> **Response to AnonReviewer4**
>
> - We have revised the manuscript to include an example of LPDR. Please have a look at Section 2.
>
> - The sentence is modified appropriately. We have changed the sentence as well the Assumption 1. Please check the revised manuscript.
>
> - Assumption 1 was modified to "expectation of random choice" and Assumption 2 was modified to "with a high probability".
>
> - Based on Theorem 1, if target $\rho_n'$ is too small, the variation ratio of all unlabeled samples becomes 0, resulting in  random query strategy. In the figure, it can be seen that the performance drops sharply when target $\rho_n'$ is less than $0.02$. It is natural to set the target Hamming distance to $q/m$ since this task has to select $q$ samples from $m$ pooling data. However, the target Hamming distance does not necessarily have to be set to $q/m$, and it was arbitrarily set in the experiments based on Figure 9.
>
> - Assumption 1 and 2 are introduced not based on or to conform to  any theory but are introduced in deriving the algorithm. Assumption 1 is introduced to sample hypotheses within the ball (near the learned hypothesis) based on sampling the parameters (or weights of a deep network), and Assumption 2 is introduced to avoid grid search (increasing perturbation strength from nil disagreement at sample point until disagreement is discovered) for comparing LPDR of samples (instead perform a perturbation and rank the variation ratio which has negative correlation with LPDR). The proposed active algorithm is derived on the premise of Assumption 1 and 2  (absolutely essential in derivation) and the theoretical concepts and empirical validations are described in detail in Section 2 and 3.
>
> - In Figure 11 (revised version), experiments for $N=1$ and $2$ are included: for $N<5$, empirical results show that hyperparameter does have have catastrophic effect on the final performance; however, for $N \ge 5$, the algorithm seems to be robust to the hyperparameter setting.  Empirical results show that for $N=5$, rather than the extreme case of $N=1$, gives sufficient performance. This can be considered as a strengths of the algorithm. From empirical results shown in Figure 9, we see that the target Hamming distance does not need to be set exactly to the value $q/m$, and if we set the $\rho_n'$ to a value to small, then we have a catastrophic drop in performance.
>
> - We have included additional experimental results to show that performance is  indeed sensitive to the hyperparameter $N$ setting. When the variance of the perturbation is either too high or too low, the variation ratio becomes close to $1$ (no majority class among sampled hypothesis prediction) or $0$ (all sampled hypothesis predicting to one common class). We tried to explain this empirical observation/phenomena as theoretically (analytically) as possible.  We tried to the best of our abilities to revise the manuscript to answer any questions and ambiguities of the reviewers.

---

### Decision · Program_Chairs · 2021-01-07
**Final Decision**

**Decision:**

Reject

**Comment:**

While the results are promising, several concerns were raised in the reviews, leading to the reject recommendation at this time.  There is an agreement among all reviewers that the paper would benefit from a revision.

Most reviewers felt that the paper lacks a rigorous and compelling theoretical justification for the proposed algorithm, making suggestions for what would make the paper stronger.

AnonReviewer4 would like to additionally convey the following message:

I would like to thank the authors to revise the statement of the assumptions according to my suggestions. However, the wording "with high probability" has rigorous mathematically meaning, so should be used with care. Their Figures still don't justify the current statement of the assumptions in my opinion.
Their experiment in Appendix E, which only contains one single example, is still too simplistic and cannot fully justify their claim. I would encourage the authors to test on more examples.